

# Supraspecific units in correlative niche modeling improves the prediction of geographic potential of biological invasions

Sandra Castaño-Quintero[1],[*], Jazmín Escobar-Luján[1],
Luis Osorio-Olvera[2], A Townsend Peterson[2], Xavier Chiappa-Carrara[1],
Enrique Martínez-Meyer[3] and Carlos Yañez-Arenas[1],[*]

[1] Facultad de Ciencias, Universidad Nacional Autónoma de México, Mérida, Yucatán, Mexico
[2] The University of Kansas, Lawrence, KS, USA
[3] Instituto de Biología, Universidad Nacional Autónoma de México, Ciudad de México, Mexico
[*] These authors contributed equally to this work.

## ABSTRACT

**Background:** Biological invasions rank among the most significant threats to biodiversity and ecosystems. Correlative ecological niche modeling is among the most frequently used tools with which to estimate potential distributions of invasive species. However, when areas accessible to the species across its native distribution do not represent the full spectrum of environmental conditions that the species can tolerate, correlative studies often underestimate fundamental niches.
**Methods:** Here, we explore the utility of supraspecific modeling units to improve the predictive ability of models focused on biological invasions. Taking into account phylogenetic relationships in correlative ecological niche models, we studied the invasion patterns of three species (*Aedes aegypti, Pterois volitans* and *Oreochromis mossambicus*).
**Results:** Use of supraspecific modeling units improved the predictive ability of correlative niche models in anticipating potential distributions of three invasive species. We demonstrated that integrating data on closely related species allowed a more complete characterization of fundamental niches. This approach could be used to model species with invasive potential but that have not yet invaded new regions.

**Submitted** 14 May 2020
**Accepted** 9 November 2020
**Published** 22 December 2020

Corresponding author
Carlos Yañez-Arenas,
carlos_yanez@ciencias.unam.mx

## INTRODUCTION

Biological invasions are considered the second most serious cause of species extinctions (*Richardson, Pyšek & Carlton, 2011*). In recent decades, invasions have become more frequent because of globalization, since humans have re-located (accidentally or intentionally) many species far outside their native geographic ranges (*Tatem & Hay, 2007*; *Wilson et al., 2009*). In general, invasive species have direct negative impacts on native species and ecosystems via predation, competition, propagation of diseases, and changes in composition of trophic webs (*Manchester & Bullock, 2000*). Eradication of invasive species

would be highly desirable before they become established, and the scientific community agrees broadly that prevention of invasions is the most effective and least expensive way to avoid negative impacts (*Miller et al., 2005*; *Thuiller et al., 2005*).

Mapping geographic regions presenting suitable environmental conditions for particular invasive species represents an important but challenging step in preventing establishment (*Bomford, 2008*). This goal implies, typically, estimating the set of abiotic conditions that allow the species to achieve positive population growth (i.e., its fundamental niche—$N_F$) (*Peterson & Soberón, 2012*). Mechanistic niche modeling represents means by which to approximate and map a species' $N_F$, since experiments under controlled conditions are carried out to estimate its physiological tolerance (*Kearney & Porter, 2004*, *2009*). However, these models require a lot of information obtained through long experimentation processes that can be quite costly (*Gallien et al., 2010*). Also, mechanistic models are usually generated in terms of one or two environmental variables, and may not capture all biologically relevant factors for the species (*Aragón, Baselga & Lobo, 2010*); interactions among variables are generally neglected; and application is limited to well-studied organisms (*Larson et al., 2014*).

Correlative ecological niche modeling (ENM) requires less information than mechanistic modeling, because it derives inferences from statistical associations between presence data and environmental dimensions. From these associations, ENMs can identify environmental conditions suitable for the species. By projecting these conditions into geographic space, one can generate hypotheses of species' potential distributions (*Peterson et al., 2011*). Correlative ENM has been used widely to study biological invasions (*Peterson, 2003*; *Thuiller et al., 2005*), but its predictive capacity has been hindered by two factors. First, it is complicated to differentiate correlation from true causality (*Dormann, 2007*), and second, the environmental diversity across the area that has been accessible historically for the species (**M**; *Soberón & Peterson, 2005*) may not be sufficient to characterize fundamental niches (i.e., the set of abiotic conditions that allow a given species to survive and reproduce in the absence of biotic interactions) fully, which reduces the $N_F$ to what is termed the existing niche: the niche that one is able to observe and characterize. Hence, if **M** does not cover the whole environmental variation that the species tolerates, ENMs end up sub-characterizing $N_F$.

This truncation of the fundamental niche imposes both conceptual and methodological challenges to anticipate the species' success of establishment when facing novel environmental conditions, such as under climatic changes and biological invasions. Therefore, when a species is environmentally restricted due to geographical constraints, for example an insular species, it is difficult to determine the portion of its niche that is represented in the island and how much is missing. Some methodological proposals to improve the predictive capacity of correlative niche models include: (1) reducing numbers of predictors to avoid environmental combinations associated with presence data being unique and statistical regularities not observable (*Peterson & Nakazawa, 2008*), (2) controlling spatial autocorrelation to decrease overfitting to calibration data (*Boria et al., 2014*), (3) balancing complexity and generalization in niche models by varying parameter settings in algorithms (*Warren & Seifert, 2011*; *Merow et al., 2014*;

*Radosavljevic & Anderson, 2014*), (4) analyzing transfer procedures in relation to strict extrapolation (*Owens et al., 2013*) and (5) using the most complete set of occurrences (presence records from both native and invaded areas) to achieve better characterization of the species' $N_F$ (*Jiménez-Valverde et al., 2011*). This latter idea improves transfers (*Escobar et al., 2016*), but its application is limited to species with populations already established outside their native ranges. A possible alternative to modeling species with **M** areas that are not sufficiently representative environmentally could be based on the idea of phylogenetic conservatism of ecological niches (PNC, *Peterson, Soberón & Sánchez-Cordero, 1999*). Although niche conservatism obviously breaks down over evolutionary time, considerable evidence indicates the frequent absence of ecological niche differentiation at time scales comparable to those of invasions and speciation events (*Peterson, 2011*). That is, niche traits are often conserved among closely related species (*Pavoine & Bonsall, 2011*). If so, one may assume that sister species have identical or very similar $N_F$'s, despite being distributed in geographic regions with different environmental characteristics. These differences in accessible environments mean that together they might be more representative of the $N_F$ of the species in a lineage. This strategy of modeling niches by grouping presence records above the species level, known as "lumping" (*Smith et al., 2018*), under certain circumstances, may be a more accurate way to characterize the niche of a lineage.

The objective of this study was to evaluate whether using supraspecific modeling units (including both the occurrences of a focal invasive species and those of sister species in their respective native ranges) can improve model capacity to predict the geographic potential of invasions. We chose three invasive species with multiple populations established outside of their native geographic ranges, which offer the advantage of abundant evaluation data. The similarity of the $N_F$'s between closely related species may vary with respect to their genetic similarity, so we evaluated model performance as we created modeling units that included records of successively less closely-related species. We observed that supraspecific units improved the prediction of geographic potential of biological invasions in the three species, which suggests that niches are highly conserved and complementary among the species that composed the unit with better predictive ability.

## MATERIALS AND METHODS

We used QGIS 3.0.2 (*QGIS Development Team, 2018*) and R 3.4.4 (*R Core Team, 2018*) to carry out all the geographic information system and statistical procedures described in the following sections.

### Presence data

We chose three species with significant invasive potential that inhabit different environments: the mosquito *Aedes aegypti* (a terrestrial species), the Mozambique tilapia *Oreochromis mossambicus* (a freshwater fish), and the red lionfish *Pterois volitans* (a marine fish). For each species, we obtained a recently published phylogeny that allowed identification of supraspecific units corresponding to successively deeper nodes on the

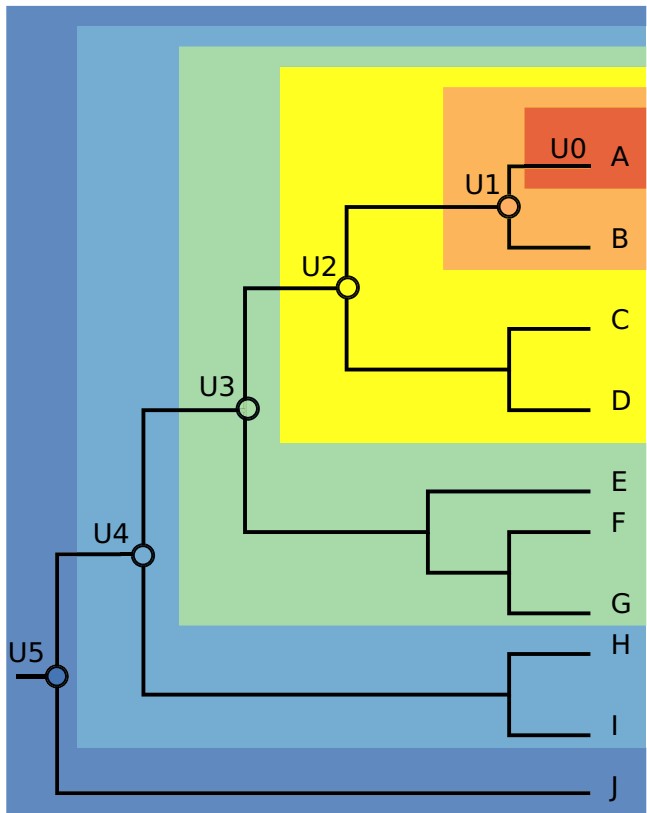

**Figure 1 Illustration of the modeling units' construction.** The first unit (U0), depicted in the dark orange area, contains occurrences from native distribution of each focal species. The second unit (U1), depicted in the light orange area, is U0 plus the native occurrences of the sister species. The next units were built by adding the native occurrences of the species that compose the following nodes, marching from the focal species down the phylogenetic tree towards its root.

phylogenetic trees. The phylogenies that we used were those reported by *Chu et al. (2018)*, *He et al. (2011)* and *Kochzius et al. (2003)*, for the three species, respectively. *Chu et al. (2018)* used a portion of mitochondrial cytochrome oxidase to infer relationships of 34 taxa of mosquitoes, *He et al. (2011)* used mitogenomic data to explore relationships among 21 tilapia species, and *Kochzius et al. (2003)* estimated relationships of seven species of the *Pterois* genus based on mitochondrial DNA sequences of 16s rDNA and cytochrome b. We built the first unit (called U0) only with data from the native distribution of each focal species. The second supraspecific unit (U1) was composed of U0 plus the data from native distribution of the sister species. We constructed and named subsequent supraspecific units similarly, marching from the focal species down the phylogenetic tree toward its root. We also built a unit (UT) including all occurrences of the focal species, native and invasive, as a more full representation of its fundamental niche (Fig. 1).

We obtained species occurrence data from the Global Biodiversity Information Facility (*GBIF, 2019*). We eliminated records without coordinates, duplicated records, and those with evident errors (e.g., records located at sea for terrestrial species or vice versa). In addition, we spatially filtered presence data by eliminating clusters of records in order to

reduce sampling bias and model overfitting (*Boria et al., 2014*) with the R package spThin (*Aiello-Lammens et al., 2015*). Number of occurrence records used for each modeling unit are shown in Table S1.

We delimited the native historical accessible areas (**M** area, sensu *Soberón & Peterson, 2005*) for each species by selecting the ecoregions that included at least one of its presence records across their native geographic range. For continental species, **M** was delimited using terrestrial ecoregions (*Olson et al., 2001*), and for marine species, it was based on the bio-regionalization of coastal areas (*Spalding et al., 2007*).

## Environmental data

We downloaded 23 global terrestrial environmental surfaces from the CliMond database (*Kriticos et al., 2012*), at a spatial resolution of 10′ (~17 km). These surfaces summarize annual trends, seasonality, and limiting environmental factors derived from monthly values of temperature, precipitation, radiation, and moisture for 1950–2000 (Table S2). To represent marine conditions, we downloaded 12 environmental surfaces from the Bio-Oracle database 2.0 (*Assis et al., 2018*). These surfaces were used at the same resolution as terrestrial ones, and represent annual patterns, seasonality and limiting factors of sea surface temperature and salinity for 2000–2014 (Table S3). We carried out a principal components analysis (PCA) to reduce multicollinearity and dimensionality using the *PCARaster* function of the R package ENMGadgets (*Barve & Barve, 2013*). We retained the first three components for both terrestrial and marine surfaces, which explained 88 and 93% of the global overall variance, respectively. The number of retained components was determined based on scree plots (terrestrial = Fig. S1; marine = Fig. S2).

## Correlative ecological niche models

We estimated ecological niches as minimum-volume ellipsoids (MVE, *Van Aelst & Rousseeuw, 2009*). We chose this approach for two main reasons. First, in this method it is assumed that the $N_F$ presents a convex shape, as suggested by theoretical and physiological evidence (*Maguire, 1973*; *Hooper et al., 2008*; *Angilletta, 2009*; *Soberón & Nakamura, 2009*). Second, by using a simple-shaped envelope method we can measure directly the contribution of including presence records of the related species than in mathematically more complex algorithms. We also modeled with Maxent to assess whether the observed patterns were method-dependent (Maxent modeling protocol and results are presented in the Supplemental Information).

We fitted MVE's in environmental space (**E**) taking into account 97.5% and 95% of the presence records. These percentages were based on the degree of confidence in the data, leaving out outlier points associated with atypical environmental conditions that could have been unidentified errors in the data-cleaning phase, or probably associated with sink population (*Osorio-Olvera et al., 2020*). We calculated both covariance matrix and centroid of MVE's using the *cov.rob* function of the MASS package through the R package ntbox (*Osorio-Olvera et al., 2016*). From U1 up to the last unit, the final ellipsoids are the result of weighting (averaging centroids and covariances) the ellipsoids built with each group of occurrences units (e.g., we built U1 by weighting ellipsoid U0 and the ellipsoid
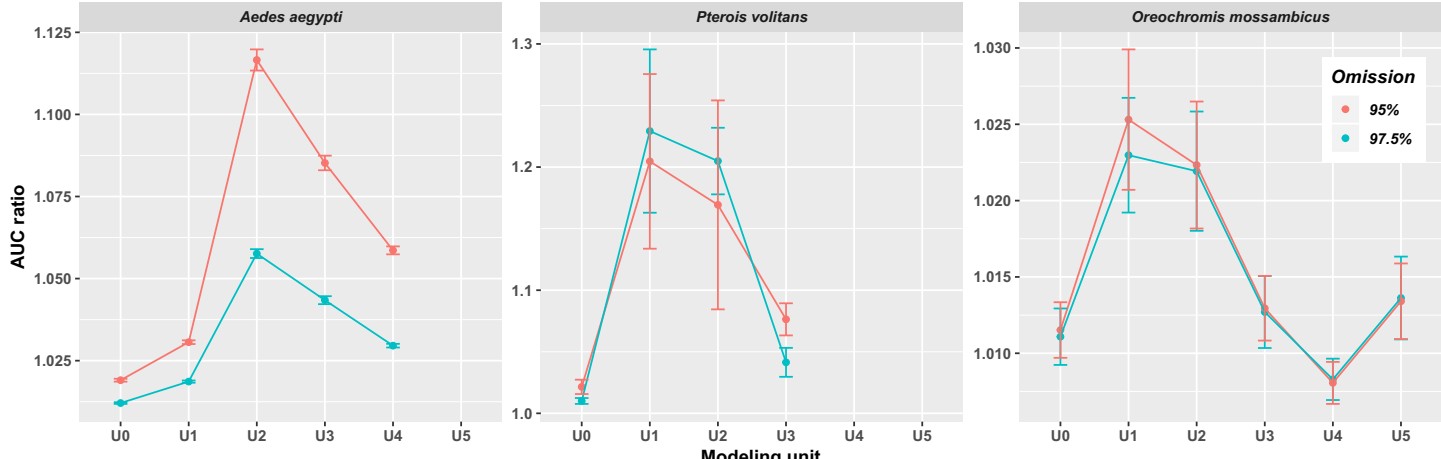

**Figure 2  Average AUC ratios of the ecological niche models obtained with the minimum-volume ellipsoid approach for each modeling unit.** Bars indicate the standard deviation.

built only with the presences of U1). To graph the MVEs, we used the R package rgl (*Adler et al., 2018*). In addition, we projected the models in geographic space as environmental suitability maps based on Mahalanobis distances from each environmental combination to the centroid of the ellipsoid (*Osorio-Olvera et al., 2016*). We estimated these distances using the R function *mahalanobis* with the ellipsoid covariance matrix, the vector of environmental variables, and the vector of means (the centroid). We converted environmental suitability models into binary maps (presence or absence of suitable conditions) using two thresholds of allowed omission: 2.5% and 5%. For visualization and manipulation of the resulting maps, we used the R package raster (*Hijmans et al., 2018*).

## Niche models evaluation

To assess performance of the ENMs, we used the AUC ratio of the partial ROC technique (*Peterson, Papeş & Soberón, 2008*). This metric evaluates the relationship between omission error for independent presence records (here, occurrences of the focal species that fell outside its native range) and the proportion of area estimated as suitable for the species, but only under conditions of low omission error (*Peterson, Papeş & Soberón, 2008*). The AUC ratio varies from 0 to 2: values greater than 1 indicate that model predictions are better than null expectations. This analysis was carried out using the R script published by *Barve (2008)*. Partial ROC test were computed based on the continuous suitability models and the presence records of the focal species from the invasion area (evaluation data); we allowed an omission error of 2.5% and 5%, and defined a bootstrapping 50% of the evaluation data 100 times to assess statistical significance of AUC ratio values.

## RESULTS

In all three species, the AUC ratio of the partial ROC test increased when occurrences of closely related species were included (Fig. 2). The increase was particularly pronounced in

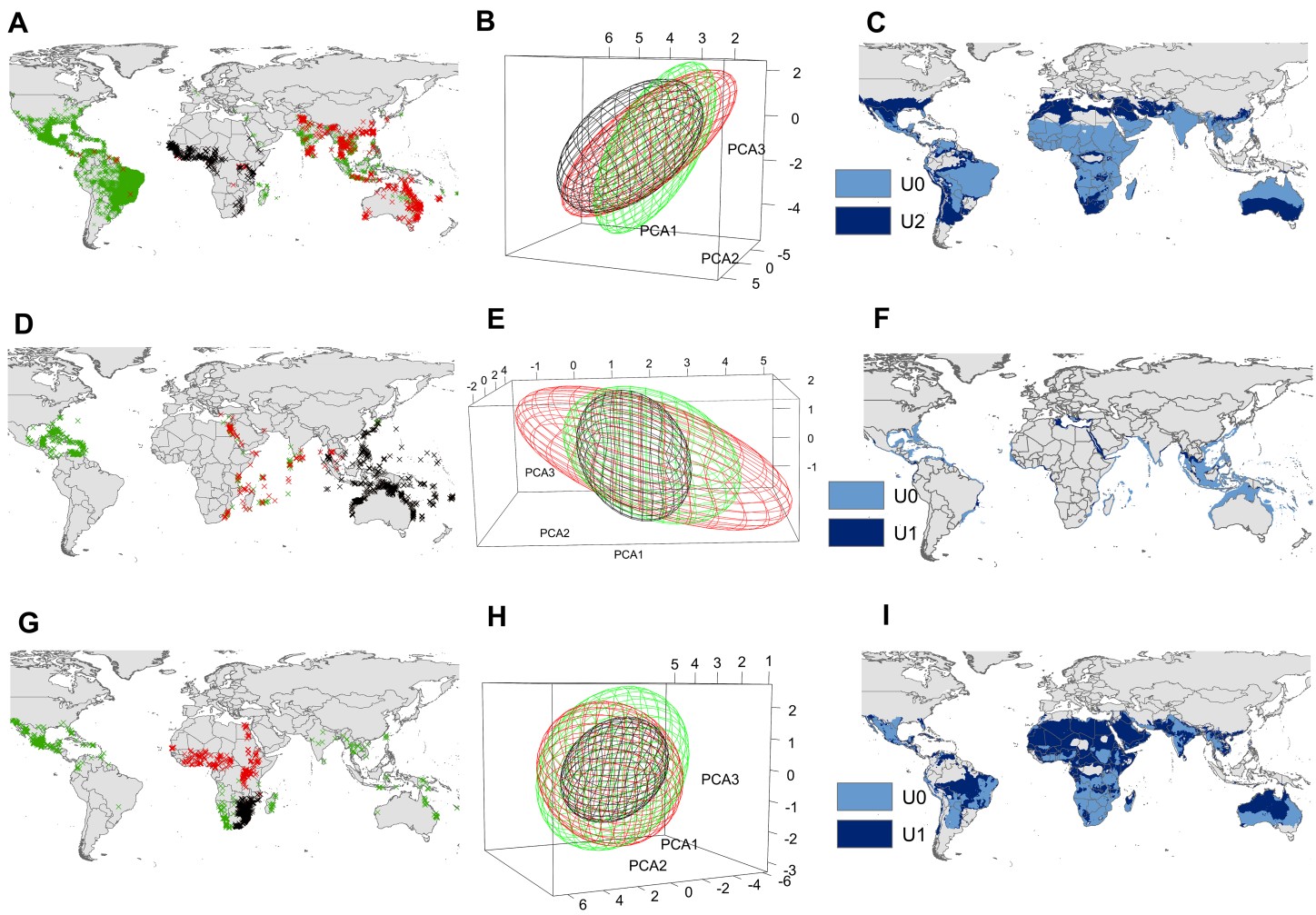

**Figure 3 Input data and ecological niche models.** Presence records of the native range (black X's), invasion range (red X's), and those included in the supraspecific unit with the highest AUC ratio (blue X's) of *Aedes aegypti* (A), *Pterois volitans* (D) and *Oreochromis mossambicus* (G). Ecological niche model constructed with the black X's (black ellipsoid) (i.e., U0), ecological niche model constructed with de black X's + blue X's (blue ellipsoid) (i.e., U2 for Ae. aegypti and U1 for *P. volitans* and *O. mossambicus*), and ecological niche model constructed with the black X's + red X's (red ellipsoid) of *Ae. aegypti* (B), *P. volitans* (E) and *O. mossambicus* (H). Potential distribution obtained from the black ellipsoid (light blue), and from the blue ellipsoid (dark blue) of *Ae. aegypti* (C), *P. volitans* (F) and *O. mossambicus* (I). Minimum-volume ellipsoids presented in this figure included 95% of each input data.

*Ae. aegypti* and *P. volitans*, whereas in *O. mossambicus* this pattern was not as conspicuous; indeed, in that species, the AUC ratio values of some supraspecific units were lower than those of U0. The highest values of AUC ratio were obtained for U1 in *P. volitans* and *O. mossambicus* and for U2 in *Ae. aegypti*. In addition, when we incorporated occurrences of successive related species, AUC ratios generally dropped, except for *O. mossambicus*, in which they remained high and even increased again in U5. Models including 97.5% and 95% of presence records were mostly very similar; however, the 95% models showed higher AUC ratios (Fig. 2), so hereafter we only show ellipsoids and thresholded maps derived from this modeling procedure.

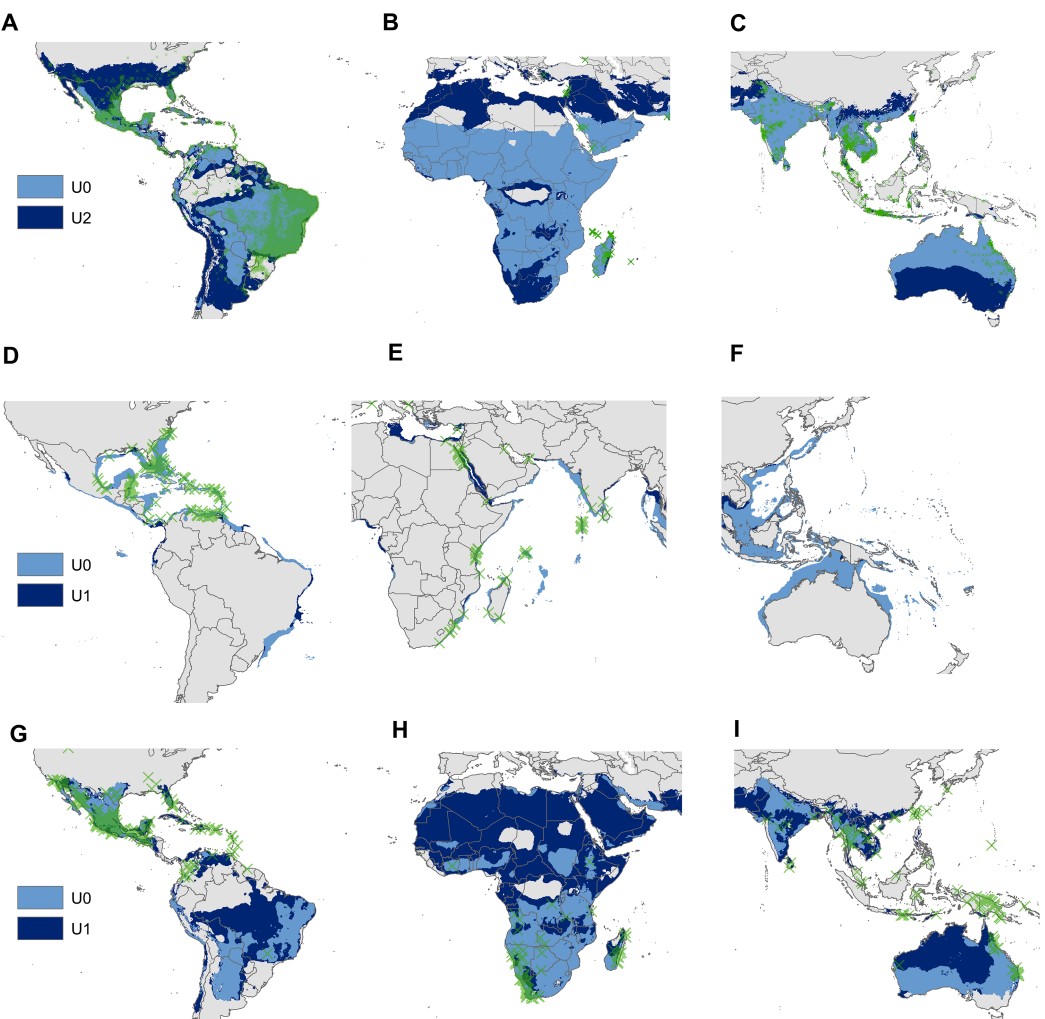

**Figure 4 Regional views of the potential distribution (obtained with the minimum-volume ellipsoids that include 95% of occurrences) across the invasion area of *Aedes aegypti* (A, B and C), *Pterois volitans* (D, E and F) and *Oreochromis mossambicus* (G, H and I).** Light blue model = distribution estimated from U0; Dark blue model = distribution estimated from the supraspecific unit with the highest AUC ratio; Green X's = presence records of the invaded areas.

The UT ellipsoid was larger than the U0 ellipsoid in *Ae. aegypti*, but smaller than that of U2 (Fig. 3B). Thresholded models of U0 failed to predict a high number of presence records outside of the species' native range. In contrast, U2 models correctly predicted many of these records (Fig. 3C), particularly in North America, some regions of South America (Fig. 4A) and Asia (Fig. 4C), the east coast of the Mediterranean Sea (Fig. 4B), and in the southeastern and southwestern coasts of Australia; although potential distribution obtained from the ellipsoid model that included 95% of presence records covered almost the entire surface of this country (Fig. 4C).

The UT ellipsoid of *P. volitans* was greater than that of U0, but smaller than U1 (Fig. 3E). Near the Bahamas and Florida, U0 and U1 predicted almost all presence records (Fig. 4D). Furthermore, U0 failed to predict some invasion records located at the

east coast of the Mediterranean Sea and along the entire coast of the Red Sea, while in these regions U1 showed high predictability (Fig. 4E).

In *O. mossambicus*, UT ellipsoid contained U0 and U1 (Fig. 3H). Distribution maps derived from the U1 modeling unit predicted invaded regions that were not predicted by U0 (Fig. 3I). For instance, in eastern Florida, central and southern Mexico, Haiti, Dominican Republic, and the Colombian Andes in the Americas (Fig. 4G); Namibia, South Africa, and central Madagascar in Africa (Fig. 4H); central India, eastern Thailand, Southwestern Indonesia and Taiwan in Asia; and northeast Australia (Fig. 4I).

## DISCUSSION

As we hypothesized, creation of supraspecific modeling units improved the predictive ability of the ENMs for the three analyzed species. Specifically, by adding the occurrences of sister taxa of each focal species (that is, when we created the U1 or U2 modeling unit), omission rates decreased considerably in the areas of invasion. This result suggests that existing niches of the species were not fully representative of their fundamental niches before invading other regions; when we incorporated presences of the most closely related species, we added relevant and complementary information for the characterization of the fundamental niches (*Godoy, Camargos & Lodi, 2018*). However, in two of the three species evaluated, predictive power decreased when additional, less closely related lineages were added, probably because these lineages have somewhat diverged in niche characteristics from the focal species. Our results are consistent with *Peterson (2011)*, who observed that niche conservatism tends to break down on temporal scales greater than those of speciation events.

In the case of *O. mossambicus*, predictive power decreased up to U4 but increased again when we added U5, which include less closely related lineages. This fluctuating result could indicate that some environments within the **M**'s of each lineage are less diverse than those contained in UT, or little ecological differentiation has occurred in this group. Previous studies have shown that ecological niches are frequently conserved in many freshwater fish clades (*McNyset, 2009*). The pattern observed in this species makes clear that, in some groups, identifying ideal modeling units may be challenging. For instance, predictive capacity in the *O. mossambicus* models reached a maximum at U1 and again went up at U5, the last unit that we evaluated, so the stability of niche characteristics has been particularly marked in this group.

Another important issue that should be taken with caution is that most of the supraspecific units overestimated the thermal tolerance limits reported in experimental studies (*Ae. aegypti* = 16.0–36.0 °C, *De Almeida et al., 2010*; *Marinho et al., 2016*; *P. volitans* = 22.0–31.0 °C, *Barker, 2015*; *O. mossambicus* = 6.0–43.3 °C, *Fast, 1986*; *Del Rio Zaragoza, Rodriguez & Buckle-Ramirez, 2008*). This may suggest that predictions outside environmental ranges associated with occurrences of the focal species could represent complementary conditions of their $N_F$'s as well as environments not suitable for the survival of their populations. Differences between the thermal ranges could be also a consequence of certain methodological limitations. For example, experimental studies

do not evaluate behavior and strategies of organisms to deal with suboptimal environments (*Kearney & Porter, 2009*; *Soberón & Arroyo-Peña, 2017*). Moreover, physiological thermal limits are usually determined based on a set of individuals that do not represent all the intraspecific variability (*Marinho et al., 2016*). On the other hand, our MVEs (constructed with an allowed omission of 2.5% and 5%) could still be including records of individuals observed in unsuitable environments but dispersed there through anthropic mechanisms (*Bruno, Stachowicz & Bertness, 2003*; *Campbell et al., 2015*) or that represent sink populations (*Pulliam, 2000*).

Despite the limitations and complications, taking into account phylogenetic relationships to carry out a "lumping" strategy (*Smith et al., 2018*) allowed us to build more informative niche models. Even if the presences of the sister species represent conditions outside the $N_F$ of the focal species, the models could be acceptable if commission errors are low (*Qiao et al., 2017*); in fact, these 'breaks' in complementarity of niches can be detected when the predictive capacity of the models decreases. However, the main reason for modeling above the species level in the case of biological invasions is to model across more environmentally diverse **M** areas, which leads to niche models that are more broadly representative and robust as regards the fundamental niche. Complementary predictions based on sister species could offer a preliminary hypothesis of areas with potential for invasion, but which should be considered after those represented in the observable, native-range niche. In a real application of "lumping", the complementary niche areas should be explicitly indicated in **G**, together with an analysis of novel environments (e.g., MOP, *Owens et al., 2013*).

Modeling above the species level opens up many possibilities for research, particularly in species with BAM configurations where full characterization of fundamental niches is likely to prove difficult; for example, "Classic" BAM or "Wallace's Dream" (*Soberón & Peterson, 2005*; *Saupe et al., 2012*). Beyond biological invasions, rigorous and systematized creation of supraspecific units in modeling ecological niches has potential to improve answers to interesting biological questions that depend on characterizing species' fundamental niches. To mention some examples, transferring models to past or future scenarios of climate change based on closely related species may yield results distinct from those in which traditional modeling approaches are used. Research on niche structure (*Martínez-Meyer et al., 2013*; *Pironon et al., 2018*) could benefit from "lumping" since the centroids of the $N_F$'s are informative about population abundance (*Martínez-Meyer et al., 2013*), population density (*Yañez-Arenas et al., 2012*), or genetic diversity (*Lira-Noriega & Manthey, 2014*).

## ACKNOWLEDGEMENTS

We thank all the students of the 'Laboratorio de Ecología Geográfica' of the Universidad Nacional Autónoma de México (particularly Lina Jiménez for their support at the initial phase of the project). To Norberto Colín for the feedback on the choice of phylogenies and to Octavio Rojas for inspiring this study with his entertaining talks about 'concepts of species and niche modeling'.

### Funding

Luis Osorio-Olvera received funding from Consejo Nacional de Ciencia y Tecnología (CONACyT; postdoctoral fellowship number 740751; CVU: 368747) and PAPIIT IN116018. The funders had no role in study design, data collection and analysis, decision to publish, or preparation of the manuscript.

### Grant Disclosures

The following grant information was disclosed by the authors:
Consejo Nacional de Ciencia y Tecnología (CONACyT): 740751 and CVU: 368747.
PAPIIT: IN116018.

### Competing Interests

The authors declare that they have no competing interests.

### Author Contributions

- Sandra Castaño-Quintero performed the experiments, analyzed the data, prepared figures and/or tables, and approved the final draft.
- Jazmín Escobar-Luján performed the experiments, analyzed the data, prepared figures and/or tables, and approved the final draft.
- Luis Osorio-Olvera performed the experiments, analyzed the data, prepared figures and/or tables, and approved the final draft.
- A. Townsend Peterson conceived and designed the experiments, prepared figures and/or tables, authored or reviewed drafts of the paper, and approved the final draft.
- Xavier Chiappa-Carrara conceived and designed the experiments, authored or reviewed drafts of the paper, and approved the final draft.
- Enrique Martínez-Meyer conceived and designed the experiments, authored or reviewed drafts of the paper, and approved the final draft.
- Carlos Yañez-Arenas conceived and designed the experiments, performed the experiments, analyzed the data, prepared figures and/or tables, authored or reviewed drafts of the paper, and approved the final draft.

### Data Availability

Data and code are available at OSF: Escobar-Luján, J. (2020, October 13). Dataset of Supraspecific units in correlative niche modeling improves the prediction of geographic potential of biological invasions. Retrieved from osf.io/7bpcg.

### Supplemental Information

Supplemental information for this article can be found online at http://dx.doi.org/10.7717/peerj.10454#supplemental-information.

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
