# Peer review of "Supraspecific units in correlative niche modeling improves the prediction of geographic potential of biological invasions"

_PeerJ, doi:10.7717/peerj.10454_

## Round 0.1 · original submission · Major Revisions

Three reviewers commented on your manuscript: their comments on your work are good, but they also pointed out a number of issues to be addressed that needs a moderate revision of the manuscript. In general, most of the comments focuses on specific clarifications needed (e.g. editing figures; giving more details on the supraspecific units, …), but some of them are more general (e.g. expanding the introduction and discussion sessions, or comparing the results with the ones obtained with more complex algorithms then the simple-shaped envelope). I feel that the manuscript can be easily edited to address these concerns, but if you think that you have to rebut any of the raised comments, please give an exhaustive explanation.

Reviewer 1 ·

Basic reporting

No comment

Experimental design

No comment

Validity of the findings

No Comment

Additional comments

Authors of this manuscript use supraspecific units for three invasive species to understand their invasive potential using correlative niche modeling. Authors explore an uncommon way of estimating ecological niches as minimum-volume ellipsoid, a kind of envelope method, than that of using a complex algorithm Maxent. The idea of modeling supraspecific units to improve the modeling potential is good. However, I have some suggestions to improve the manuscript.

1. Instead of using only envelope methods, it would be interesting to compare the similar modeling framework with complex modeling algorithms such as maxent.

2. Authors perform models for supraspecific entities such as U0, U1, U2 and UT. But the explanation only written in text is not very clear, If you can explain these in some cartoon figures, it will be easier for the reader. Because later, in the manuscript, in the results section, there is sudden mention of entity U5, which was not clearly mentioned earlier in the manuscript.

3. Figures need a lot of improvement. Map legend is almost not readable in figure 2. Also the black dots and green dots are not visible in the map. Also, figures presented in the manuscript are only of one species. I was expecting all the species figures in the main manuscript. Because part of the discussion section is devoted to comparing how this modeling framework differs based on species. But to refer to these, one has to go to supplementary material

4. Also there is no mention of how many occurrences were used for each supraspecific modeling entity.
You can certainly add a table for such details.

Reviewer 2 ·

Basic reporting

The manuscript is well-written; the literature is appropriated and timely; the submission is 'self-contained'.

Experimental design

It is an original primary research, well defined, relevant and meaningful. Methods are well described.

Validity of the findings

Underlying data were provided; conclusions are linked to the aims.

Additional comments

I am very happy by invitation to review the manuscript by Castaño-Quintero et al., that proposes using data from closely related species to better characterize fundamental niche of a given focal species and to investigate potential for biological invasions. The results support the proposition that, based on the assumption of ecological niche conservatism, integrating data from sister species (or from deeper evolutionary units) in niche modeling can improve predictions for invasive species. No doubt, this is a relevant and timely research issue, which approach can be applied to anticipate biological invasions and prevent their consequences. The text is well-written, and the methods are very robust to test their aims. I have only minor comments related to some points that I think should be clearer.

1. May be it will be interesting to indicate (in the supplementary material) which are the supraspecific entities for each species U0; that is, which are the U1, U2, …I only saw how many species there were in the Figure 1, but you could include a table (for instance) with information of these species and the number of occurrence records for these entities in supplementary material.
2. Some objective explanation for retaining only the first three components from PCA? In the case of terrestrial variables, they only explained 88% of the variance. Why not investigate scree plots or apply Kaiser’s criterion? I am not sure (don’t remember now), but I think the NMTML package from De Andrade et al., 2020 you cited have some approach to select PCA components.
3. Resolution of some figures should be improved. Figure 2, for instance, I couldn’t see the occurrence records.
4. Page 31, complete the sentence “… and the green dots where the species.?”. See line 547.

Reviewer 3 ·

Basic reporting

This manuscript is well written, easy to read, and presented clearly with a few exceptions (which are noted below). It is very concise, and would benefit from expanding both the Introduction and the Discussion a bit, in order to help situate their contribution in both applied and theoretical ecological contexts.

All the maps in the main document are difficult to read. I recommend enlarging them and/or trying different color schemes with higher contrast.
Figure 2. The legends on A and C , and the axes on B and D are too small to be legible.

Experimental design

The methods used are sound. The only aspect I question is , where is the evidence that niche traits are conserved among the “closely related” species used in this study? I think that should be clearly stated since this underlies the premise that is fundamental to their study, namely that niche traits are closely conserved, thus the addition of sister species should improve/expand the delineation of the fundamental niche and improve modeling of the realized niche. Also, while descriptive data and R code to reproduce the PCAs are included, but raw data (or any data) used to make the models are not.

Validity of the findings

no comment

Additional comments

This manuscript is well written, easy to read, and presented clearly with a few exceptions (which are noted below). It is very concise, and would benefit from expanding both the Introduction and the Discussion a bit, in order to help situate their contribution in both applied and theoretical ecological contexts. For example, in the Abstract the authors state that “This approach could be used to model species with invasive potential but that have not yet invaded new regions.” I agree with this and it is a pity that they do not make more explicit ties to this idea in their paper (especially in the introduction and Discussion). This paper would be considerably strengthened if they spend more time on contextualizing their research in the introduction with respect to 1) the theoretical and methodological problem of realized niches being used in ENMs to delineate the fundamental niche, and, 2) how this leads to the failure to predict alien species invasions or the full extent of alien species distributions which in turn has negative ramification for early detection and eradication.
The methods used are sound. The only aspect I question is , where is the evidence that niche traits are conserved among the “closely related” species used in this study? I think that should be clearly stated since this underlies the premise that is fundamental to their study, namely that niche traits are closely conserved, thus the addition of sister species should improve/expand the delineation of the fundamental niche and improve modeling of the realized niche. Also, while descriptive data and R code to reproduce the PCAs are included, but raw data (or any data) used to make the models are not.

I list some specific points below:
L 71-72 Clearly Define fundamental niche for readers unfamiliar with the concept. It is already stated here implicitly. Also, it would be more consistent to refer to “existing” niche as the “realized” niche which is the standard term in the literature. (e.g. Soberon & Peterson, 2005, Barve et al., 2011)
L117-119- it would be helpful to briefly describe how these phylogenies were obtained
L133 I would rephrase this. You are not testing hypotheses of which areas have been historically accessible, rather you are making an assumption (which I am not arguing with) that ecoregions represent historically accessible areas to the organisms.
166-169- it is not clear how these atypical environmental conditions/sink populations are identified, please elaborate. A citation would be nice.
Line 169- please elaborate what the cov. rob function does
Lines 171 and 172- do you mean “weighting”? The use of the word “weighing” does not make sense in the context presented.
L197-199- this is unclear
236-243. This is plausible. However, how do we know the findings are not simply a statistical artefact of the sister species occupying the same geographic areas as the focal species?

General comment- you can state “all statistical analyses were carried out in R3.4.4, once in the manuscript.
All the maps in the main document are difficult to read. I recommend enlarging them and/or trying different color schemes with higher contrast.
Figure 2. The legends on A and C , and the axes on B and D are too small to be legible.

References:
SOBERÓN, JORGE, and A. TOWNSEND PETERSON. "INTERPRETATION OF MODELS OF FUNDAMENTAL ECOLOGICAL NICHES AND SPECIES’DISTRIBUTIONAL AREAS." Biodiversity Informatics 2 (2005): 1-10.
Barve, N., Barve, V., Jiménez-Valverde, A., Lira-Noriega, A., Maher, S. P., Peterson, A. T., ... & Villalobos, F. (2011). The crucial role of the accessible area in ecological niche modeling and species distribution modeling. Ecological Modelling, 222(11), 1810-1819.

---

## Round 0.2 · accepted · Accept

Two of the reviewer re-reviewed your manuscript finding that your responses to be satisfactory. I agree that the modifications proposed improved the manuscript and answered all the issues raised in the previous round of reviews. Indeed, I think that this work represents a very interesting contribution to the field of species distribution models.

Reviewer 2 ·

Basic reporting

no comment

Experimental design

no comment

Validity of the findings

no comment

Additional comments

This version of the manuscript is greatly improved. The authors included additional information in the Materials and Methods and in the supplementary material, which make the manuscript much clearer; changes done in the Introduction and in the Discussion provided a coherent narrative throughout the text. The Figures were also improved. I have no more suggestions to do.

Reviewer 3 ·

Basic reporting

The authors addressed all my comments satisfactorily.

Experimental design

no comment.

Validity of the findings

no comment

Additional comments

Congratulations on your manuscript! It will be a nice contribution to species distribution modelling.